# Disease Modeling of Rare Neurological Disorders in Zebrafish

**DOI:** 10.3390/ijms23073946

**Published:** 2022-04-01

**Authors:** Myeongjoo Son, Dae Yu Kim, Cheol-Hee Kim

**Affiliations:** 1Inha Research Institute for Aerospace Medicine, Inha University, Incheon 22212, Korea; mjson@inha.ac.kr; 2Center for Sensor Systems, Inha University, Incheon 22212, Korea; 3Department of Electrical Engineering, College of Engineering, Inha University, Incheon 22212, Korea; 4Department of Biology, Chungnam National University, Daejeon 34134, Korea

**Keywords:** rare diseases, neurological rare diseases, undiagnosed diseases, model organisms, *Danio rerio*, zebrafish

## Abstract

Rare diseases are those which affect a small number of people compared to the general population. However, many patients with a rare disease remain undiagnosed, and a large majority of rare diseases still have no form of viable treatment. Approximately 40% of rare diseases include neurologic and neurodevelopmental disorders. In order to understand the characteristics of rare neurological disorders and identify causative genes, various model organisms have been utilized extensively. In this review, the characteristics of model organisms, such as roundworms, fruit flies, and zebrafish, are examined, with an emphasis on zebrafish disease modeling in rare neurological disorders.

## 1. Introduction

Rare diseases affect a relatively small portion of people compared to other prevalent diseases; thus, characteristic issues arise, due to their rarity [1]. Although the definition of a rare disease varies slightly from country to country, the United States Congress defined a rare disease in the Orphan Drug Act of 1983 as a condition affecting fewer than 200,000 patients. The total number of Americans suffering from rare diseases is estimated at 25–30 million, while 30 million people are affected by rare disease across Europe [1].

Globally, there are about 8000 rare diseases, including genetic disorders, rare cancers, auto-immune disease, and infectious diseases, which are often serious, progressive, and chronic conditions [2]. Many rare diseases show a variety of signs starting at birth or childhood, including proximal spinal muscular atrophy, neurofibromatosis, chondrodysplasia osteogenesis imperfecta, and Rett syndrome. Rare diseases which appear during adulthood include those such as Huntington diseases, Crohn diseases, amyotrophic lateral sclerosis, and Charcot–Marie–Tooth diseases.

The exact cause of many rare diseases is still unknown, but a large majority of them (~80%) have a genetic cause, including a direct single gene, multifactorial, or chromosome changes [3]. In some cases, genetic changes causing disease are passed from one generation to the next. In other cases, they randomly occur in a person who is the first in a family to receive diagnosis [3]. Many patients with rare conditions undergo a “diagnostic odyssey”, trying various clinical approaches and comprehensive biochemical and genetic tests in hopes of an accurate assessment; however, often, they must wait years for a definitive diagnosis. National and international researchers have made progress in learning how to diagnose, clinically treat, and even prevent many rare diseases (Table 1). Unfortunately, 30% of child patients, which make up 50% of rare-disease patients, die before the age of 5 [3], while approximately 40% of known rare diseases include neurologic and neurodevelopmental disorders.

In this article, various model organisms developed or utilized to characterize rare diseases for diagnosis and research are addressed. In addition, the trends of rare neurological disease research, using zebrafish models and potential therapeutic developments, are discussed.

## 2. Model Organisms for Rare Disease Research

In vitro approaches that use mainly cell or tissue culture can help predict clinical outcomes [4] but are limited in mimicking rare human diseases. The selection of an appropriate model organism is critical for preclinical research. Several important factors of consideration include species similarity to humans (i.e., the closer the phylogeny, the more similar the genetic composition, anatomy, and physiology), genetic homogeneity, a priori knowledge, cost, availability, translatability of results, ease of operation, ethical implications, etc. [5].

Over the past few years, clustered regularly interspaced repeat (CRISPR)/CRISPR-associated protein 9 (Cas9) system (CRISPR/Cas9) genome editing technology has transformed the field and has greatly expanded the repertoire of cell/animal systems available for rare disease modeling [6]. The use of this genome editing technology with genetic model organisms has enhanced our understanding of human rare diseases. For example, mouse and zebrafish models, at first glance, appear completely unrelated to humans; however, on the genetic and physiological level, they respectively share about 85% and 71% of the same genes and possess major organ systems in common, such as the central nervous system, circulatory system, digestive system, etc. To study the functional consequences of the hundreds of rare variants discovered by genome sequencing, researchers have developed the use of specific model organisms, including roundworms (*C. elegans*), fruit flies (*Drosophila melanogaster*), and zebrafish (*Danio rerio*).

### 2.1. Caenorhabditis elegans (C. elegans)

The significance of non-mammalian model organisms has been recognized for quite some time [1,6]. The Nobel Prize in Physiology or Medicine has been awarded to researchers for their discoveries in apoptosis (2022) and RNA interference using worms (2006).

*C. elegans* is an unsegmented pseudocoelomate, lacking circulatory and respiratory systems, whose body segments are serially repeated one after the other [7]. *C. elegans* was used primarily for neuronal development research in 1963 and has since seen extensive use as a model organism for research into neural and molecular mechanisms of learning, memory, mating behavior, chemotaxis, thermotaxis, and mechano-transduction [8]. In addition, the *C. elegans* model has provided insights into finer mechanistic details of human health, including inter-cellular signaling pathways (e.g., Notch signaling), intra-cellular pathways (e.g., autophagy), molecular machines (e.g., spliceosome), and multi-cellular processes (e.g., basement membrane biology) [9].

In 2019, the first *C. elegans* multicellular organism underwent whole-genome sequencing and is the only organism to have completed its connectome (the “wiring diagram” of neurons) [10,11,12]. There are 20,512 protein-coding genes [13], and 83% of the worm proteome is found to have human homologous genes [14]. Only 11% or less contain roundworm-specific genes. These findings provide the basis by which *C. elegans* has served as a suitable model organism for human gene functional research [15,16,17,18]. In the case of rare disease modeling, *C. elegans* provides an ideal system to study human diseases.

### 2.2. Fruit Fly (Drosophila melanogaster)

*Drosophila melanogaster* is a species of fly in the Drosophila family (taxonomic order Diptera). For the past century, the *Drosophila melanogaster* has been used as a model organism to understand the fundamentals of genetics, developmental biology, immunity, and neuroscience [19,20]. Five Nobel Prizes have been awarded to fruit fly scientists for their work with the animal in 2017. *Drosophila melanogaster* has emerged as an important model system for dissecting and understanding the molecular mechanisms underlying rare human diseases, due to its rapid life cycle, relatively simple genetics (only four pairs of chromosomes), and large number of offspring per generation. This rise is owed in part to the first genome-wide survey of ~1000 genes registered in the Online Mendelian Inheritance in Man (OMIM), which found that 75% of disease-causing genes in humans are conserved in Drosophila [21]. Of the approximately 4000 human disease-related genes currently displayed in OMIM, ~85% have homologues in Drosophila. Considering that ~65% of protein-coding genes are conserved between humans and flies, the data suggest that genes conserved between these species are more likely to implicate genetic disease in humans [22,23]. In addition to being used as a tool for dissecting rare disease mechanisms and exploring potential therapeutic avenues, flies have emerged as a key tool in explaining variants of uncertain significance found in patients [24,25].

### 2.3. Zebrafish (Danio rerio)

The zebrafish (*Danio rerio*) is a small freshwater fish, 3-to-4 cm long, with a lifespan of about 2 years. Zebrafish have become increasingly important in scientific research since the 1960s, due to several distinct advantages over other vertebrate models.

Because of the simplicity of their natural habitat, lab maintenance of a zebrafish colony is much easier than simulating the conditions necessary for mammals. Therefore, zebrafish can be grown in a cost-effective manner. Their short generation time of 3 months helps accelerate experimental progress [26], and ex utero development facilitates the observation and rapid experimental manipulation of embryos. In addition, zebrafish have large clutch sizes, ranging from 200 to 300 embryos per adult mating pair, which ensures a robust stock of animals for research work. Due to these features, combined with the relatively small size of the embryo/larva/adult, zebrafish are well-suited for high-throughput screening of potential neuroactive compounds.

The zebrafish possesses many characteristics that make it an invaluable model to study human diseases [27]; however, one of its unique advantages is the unparalleled optical clarity of the embryo, which allows for the visualization of individual genes (fluorescently stained or labeled) throughout development, using non-invasive imaging techniques [28,29,30,31] (Figure 1A). This transparency of the embryo also facilitates genetic manipulation, such that gene function can readily be studied by the injection of synthetic mRNA or plasmid DNA into early stage zebrafish embryos generating transgenic zebrafish lines or altering gene function through genome editing techniques, such as the inclusion of zinc finger nucleases (ZFNs), transcription activation-like effector nucleases (TALENs), and the clustered regularly interspaced repeat (CRISPR)/CRISPR-associated protein 9 (Cas9) system [32,33,34]. Furthermore, the zebrafish genome has been sequenced, and 71% of human genes and 82% of human disease-related genes have orthologs in zebrafish [35]. Both ZFNs and TALENs require the creation of customized protein compositions for each target site, and these systems are not suitable for large-scale applications. However, the CRISPR/Cas9 system relies on target-site recognition by a custom guide RNA (gRNA) molecule and requires only one oligonucleotide to be designed for each target site. The success of CRISPR-mediated transgenic transformation in zebrafish can largely be attributed to frame-shift-generating null alleles via non-homologous extremity joint (NHEJ)-mediated CRISPR-induced DNA break repair [35].

Interestingly, similarities between the zebrafish and human nervous system (anatomy and physiological signaling) have been reported [33]. The zebrafish brain is composed of the forebrain (telencephalon), midbrain (optic tectum), and hindbrain (cerebellum) (Figure 1B); and many cells, including astrocytes, oligodendrocytes, microglia, cerebellar Purkinje cells, myelin, and motor neurons, are also similar to human cells. Further studies of spinal nerve patterning, neural differentiation, and vertebrate network development in adult zebrafish revealed similarities to higher vertebrates [33]. Due to these characteristics, zebrafish are widely used to validate candidate disease genes and to elucidate the molecular mechanisms and pathophysiology of neurological disease. The continued increase in the use of zebrafish in biomedical research publications reflects their expanding popularity [33] (Table 2).

## 3. Zebrafish Models for Rare Neurological Disorders

### 3.1. Kallmann Syndrome (WDR11)

Kallmann syndrome (KS, OMIM #308700, #147950, #244200, #610628, #612370, and #612702) is an inherited disorder that prevents a person from beginning or fully completing puberty. KS is a form of a group of disorders called idiopathic hypogonadotropic hypogonadism (IHH, OMIM #146110), which exhibits impaired sense of smell, which is thought to be due to the developmental failure in the migration of gonadotropin-releasing hormone neurons along olfactory axonal projections [42]. In the past two decades, researchers have found that many candidate genes, including KISS1R (OMIM #604161) encodes GPR54, TAC3 (OMIM #162330), TACR3 (OMIM #162332), NELF (OMIM #608137), CHD7(KAL5) (OMIM #608892), FGF8(KAL6) (OMIM #600483), and GNRH1 (OMIM #152760); PROKR2(KAL3) (OMIM #607123) and PROK2 (KAL4) (OMIM #607002); and KISS1R. However, the genetic cause remains unknown in about two-thirds of patients with IHH and KS.

Kim and colleagues found that six patients exhibited a total of five different heterozygous missense mutations in WDR11, a gene involved in human puberty. These included three alterations in the WD domain important for beta-propeller formation and protein–protein interaction (A435T, R448Q, and H690Q). In addition, EMX1, a homeobox transcription factor related with cell fates in the developing central nerve system (CNS) [50], was found to interact with WDR11. In the zebrafish model, *wdr11* and *emx1* expression and their protein interactions were suggested during early CNS development [42].

### 3.2. Potocki–Shaffer Syndrome (PHF21A)

Potocki–Shaffer syndrome (PSS, OMIM #601224) is a rare contiguous genetic disorder with various distinguishing features, such as skeletal anomalies, eye abnormalities, multiple exostoses, craniofacial anomalies (CFA), and intellectual disability (ID) [51,52,53,54,55,56]. These phenotypes are caused by interstitial deletion of the p11.2 band of chromosome 11. The genes responsible for the last two phenotypes of this chromosomal region have been identified: deletion of *EXT2*, leading to multiple exostoses [57], and deletion of *ALX4*, leading to parietal pores [51,58]. Although a single gene was found to be responsible for multiple exostoses and parietal foramen in PSS, the phenotypes of related genes ID and CFA remain elusive.

Kim and colleagues found that the disrupted translocations of *PHF21A* at 11p11.2 associated with abnormal craniofacial and intellectual development added to the evidence that regulation of gene expression through chromatin modifications is critical for both processes [45]. To directly test the developmental importance of *PHF21A*, they isolated the zebrafish *phf21a* ortholog, examined its expression pattern, and performed loss- and gain-of-function experiments, using a zebrafish model [45]. In addition, a disruption of *PHF21A* was shown to be involved in diverse neurological phenotypes, including epilepsy, hypotonia, and neurobehavioral problems in autism spectrum disorder (ASD) [59].

### 3.3. Miles–Carpenter Syndrome (ZC4H2)

Miles–Carpenter syndrome (MCS) is an X-linked intellectual disability (XLID, OMIM# 312840) syndrome that was first described in 1991 [60]. The syndrome is characterized by short stature in men, exotropia, microcephaly, pinched fingers, long hands, wagging feet, spasticity, and severe intellectual disability. Males who harbor variants on the X-chromosome account for approximately two in 1000 males with ID. Efforts by the global research team have identified 145 XLID genes responsible for 63 non-syndromic XLID and 114 XLID syndrome entities. Exome sequencing has helped mutational analysis of the coding region of the X-chromosome and has identified 28 of 145 XLID genes over the past decade. Despite these achievements, more than 33 non-syndromic XLID and 56 XLID syndrome entities remain [61].

May and colleagues identified four mutations in *ZC4H2* at *Xq11.2*, one in-frame insertion and three missense (p.R18K, p.R213W, and p.V75in15aa) in the original family with MCS and in three other families with microcephaly, spasticity, short stature, contracture, epilepsy, and ID. In the homozygous *zc4h2* knockout mutation zebrafish model, larvae exhibited abnormal swimming, increased twitching, motor hyperactivity, eye-movement deficits, and pectoral fin contractures. These phenotypes were reminiscent of human patients with *ZC4H2* mutation. This suggests that the *zc4h2* zebrafish model could be utilized to determine the underlying cellular mechanisms of interneuronopathy and movement disorders [44].

### 3.4. The 12q24.31 Microdeletion Syndrome (KDM2B)

Microdeletion syndrome (submicroscopic deletions) is a rare chromosome disorder which exhibits chromosomal deletions (~5 Mb), including several genes which are too small to be karyotyped. This syndrome is the frequent cause of neuropsychiatric disorders that lead to ID, as well as autistic features accompanied by epilepsy and CFA. There are cases of 12q24.31 microdeletions, whose clinical features include developmental delay, epilepsy, seizures, spasms, CFA, tapering fingers, hypotonia, autistic behavior, ID, and speech delay [62].

Researchers began conducting microarray analyses to confirm deletion mapping of copy number variations at 12q24.31 of patients, resulting in seven genes, one microRNA, and one non-coding gene identified. Using whole-mount in situ hybridization analysis of the zebrafish model, tissue-specific expression levels of candidate genes involved in the 12q24.31 microdeletion were validated. Among the candidates, *kdm2bb* was extensively expressed in the midbrain, hindbrain, forebrain, and retinal photoreceptor cell layer of the zebrafish model. Kdm2b is required for proper closure of the neural tube and optic fissure. It plays an important role in neural development, and Kdm2b deficiency leads to increased cell proliferation and cell death in neural progenitor cells [63].

Unfortunately, the findings did not confirm phenotype and behavior changes of *kdm2bb* mutant zebrafish in this study [62]. However, homozygous *Kdm2b* knockout mice exhibited neural tube closure failure during embryonic neurodevelopment, due to excessive apoptosis of neuroepithelial and mesenchymal cells, resulting in extracerebral malformations and death shortly after birth [62,63].

### 3.5. Down Syndrome and Autism (DYRK1A)

Dual Specificity Tyrosine Phosphorylation Regulated Kinase (DYRK)1A (DYRK1A) is a member of the DYRK family, and *DYRK1A* is known to play a critical role in cell proliferation, differentiation, and survival during neurogenesis [64]. *DYRK1A* is located on the distal terminal of the long arm of chromosome 21 related to the “Down syndrome critical region” (DSCR). *DYRK1A* mutations have demonstrated features associated with primary microcephaly, ID, and ASD [65,66,67,68].

To understand the molecular mechanisms underlying microcephaly and ASD, researchers generated a *dyrk1aa* (7 bp deletion) mutant zebrafish model by TALEN technology. During larval stage, *dyrk1aa* mutant zebrafish showed no changes of neural stem cell marker, *sox2*, or neuronal determination marker, *neurog1*, while similar patterns of response to visual stimuli were observed between the wild type and mutant. In adult mutant zebrafish, body length and overall morphology were not significantly different compared to wild type (WT), but smaller brain size was confirmed upon dissection. Furthermore, the mutant zebrafish showed anxiolytic behavior and impaired social interaction and social cohesion in a battery of social behavioral assays [41].

### 3.6. The 12q14.1 Deletion Syndrome (SAM2)

Emotional responses, such as fear and anxiety, are a fundamental behavioral phenomenon relating to strong fitness in all species [69]. These responses are modulated by various neuro-modulators and the habenula (Hb), which is the brain region associated with the subpallium and hypothalamus, responsible for the regulation of addiction and mood disorders, including fear and anxiety [70,71]. Emotional dysregulation can lead to severe behavioral problems and impaired social interaction associated with a variety of disorders (bipolar disorder, attention deficit hyperactivity disorder, and post-traumatic stress disorder) [72,73]. For this reason, research into the genetic and biological mechanisms of emotional dysregulation is significant.

A chemokine-like gene family, called *samdori* (*sam*), was discovered in connection with emotional regulation. As a member of *sam*, *sam2* exhibited unique brain-specific expression, mainly in the dorsal Hb (dHb) and neurons in the telencephalic region and hypothalamus. Researchers developed a *sam2* mutant zebrafish and validated fear and anxiety behaviors, including thigmotaxis, freezing, or erratic movement. Researchers also found that purified SAM2 protein increased inhibitory postsynaptic transmission to corticotropin-releasing hormone (CRH) neurons in the paraventricular nucleus, which is involved in stress and anxiety responses. In addition, they identified a human homologue of *SAM2* and were able to refine a candidate gene region containing *SAM2* among 21 annotated genes which were associated with intellectual disability and autism spectrum disorder in 12q14.1 deletion syndrome [49].

### 3.7. Armfield XLID Syndrome (FAM50A)

There are 20–30% more males than females in the ID population, due to the enrichment of genes of XLID disorders caused by hemizygous variants having a significant impact on the male ID population [74].

Armfield XLID syndrome, first reported in 1999, is characterized by impaired growth and results in deformities of facial features (ocular abnormalities), postnatal growth retardation (prominent forehead and variable head circumference), and seizures [75]. Researchers reported causal variant segregating from the Armfield syndrome phenotype. As part of an XLID gene screen targeting Xq28, they identified an ultra-rare FAM50A (family with sequence similarity 50 member A; termed XAP5 or HXC26) in affected males and unaffected female carrier missense variants. The researchers utilized the zebrafish model to investigate FAM50A function, establish correlation with the Armfield XLID clinical spectrum, and test variant pathogenicity [40]. Furthermore, *fam50a* was knocked out to study its effects on early development, and it was found that mutant zebrafish exhibited physical abnormalities similar to those in humans with Armfield XLID syndrome. Affected males with FAM50A mutations exhibited dysmorphic facial features, including anterior–posterior shortening of the pharyngeal skeleton with delayed branchial arch patterning. In addition, the *fam50a* KO zebrafish model was also used for Guion–Almeida type mandibulofacial dysostosis caused by mutations in *EFTUD2* (B and C complexes) [76].

### 3.8. Leukodystrophy (Hypomyelination)

Dysmyelinating diseases, or leukodystrophy, affect white matter, the protective covering of nerve cells, the brain, spinal cord, and peripheral nerves (myelin). The word “leukodystrophy” is derived from “leuko”, meaning “white”, and “dystrophy”, denoting “abnormal growth”. Leukodystrophy is caused by inherited enzyme deficiency, destruction, abnormal formation, or renewal of myelin sheaths. There are more than 50 types of leukodystrophy. Some types are present at birth, while others may not cause symptoms until the infant reaches childhood. Some types affect mainly adults. Most types deteriorate over time; thus, symptoms of leukodystrophy can gradually lead to loss of behavior, walking, speech, hearing, muscle tone, balance, mobility, and the ability to eat.

#### 3.8.1. Leukodystrophy with Vanishing White Matter (VWM)/Childhood Ataxia with CNS Hypomyelination (CACH)

Vanishing white matter (VWM, OMIM #603896), also known as childhood ataxia with CNS hypomyelination (CACH), is one of the most common leukodystrophies [77]; however, its exact incidence has not been determined. It mainly affects children, but it can occur at all ages, from birth to adulthood. It is dominated by cerebellar ataxia and is susceptible to stress, which is a factor in the onset of disease or rapid deterioration of neurological function, which can lead to death.

VWM is caused by mutations in one of five genes, namely *EIF2B1, EIF2B2, EIF2B3, EIF2B4, and EIF2B5*, which encode five subunits of a eukaryotic translation initiation factor 2B (eIF2B) protein. This protein is necessary to produce all other proteins in the body and for regulating the rate of protein production. Despite the ubiquity of eIF2B, VWM is characterized by leukodystrophy, cystic degeneration, astrogliosis without glial scar, increased white matter sparsity and malformation of immature astrocytes, oligodendroglial progenitor cell increase, and failure to mature into myelin-forming cells [78,79,80,81,82,83,84].

Recently, researchers established the zebrafish model of EIF2B subunits, including *EIF2B3*. *EIF2B3* is required for myelination early in CNS development. The *EIF2B3* mutant zebrafish was generated by CRISPR mutagenesis and exhibits key human phenotypes including defected myelin gene expression and glial cell differentiation. Furthermore, novel *EIF2B3* variants (L168P) have been identified in a Korean patient with VWM phenotypes [48].

#### 3.8.2. Charcot–Marie–Tooth Diseases (CMT)

Primary inherited motor-sensory neuropathies are collectively known as Charcot–Marie–Tooth disease (CMT), a familial slowly progressive peroneal muscular atrophy described by three researchers in 1886 [85]. CMT can be classified into five types according to pathology and phenotype of motor and sensory neurons: type 1 (demyelinating), type 2 (distal axonal degeneration), intermediate form (myelinopathy and axonal), type 4 (recessive demyelinating), and type X (axonal degeneration with myelin abnormalities) [86]. More than 90% of CMT patients have mutations in the peripheral myelin protein 22 (*PMP22*), myelin protein zero (*MPZ*), mitofusin 2 gene (*MFN2*), or gap junction protein beta 1 (*GJB1*) genes [87]. For the remaining 10%, inheritance patterns and associated comorbidities can help determine genetic etiology.

Researchers generated a zebrafish model by injecting zebrafish with constructs harboring human *PMP22*, and they confirmed *PMP22* expression in Schwann cells along motor nerves [88].

### 3.9. Amyotrophic Lateral Sclerosis (Lou Gehrig’s Disease)

Amyotrophic lateral sclerosis (ALS) is a progressive neurological disease that affects nerve cells in the brain and spinal cord, causing the loss of muscle control. ALS is often referred to as Lou Gehrig’s disease, named after the baseball player who was diagnosed with the disease. ALS usually begins with a loss of muscle strength and gradually leads to paralysis. Ultimately, most patients die of frailty within two to five years of diagnosis [89,90]. Most ALS cases are sporadic (sALS), but about 10% are familial (fALS) and have a strong genetic link [91]. Over the past two decades, several mutations in more than two dozen genes have been identified in ALS, including superoxide dismutase 1 (*SOD1*), fused in sarcoma (*FUS*), *TARDBP* (coding for the protein TDP-43), and *C9ORF72*. Of these, about 5–10% are caused by mutations in the FUS gene on chromosome 16 [92,93] or *TARDBP* gene on chromosome 1 [94,95,96,97]. Although these genes account for only a small fraction of fALS cases, their gene products appear to be involved in the pathogenesis of most ALS cases with sALS, while the related disease frontotemporal lobar degeneration (FTLD-U) shows an association with ubiquitin-positive inclusions.

One of the first genes, *SOD1*, was validated with a subset of patients in 1993 [6]. The SOD1 gene is responsible for 20% of fALS cases [98,99,100]. More than 150 *SOD1* mutations have been identified [101,102]. Researchers generated transgenic zebrafish that expressed intermediate levels of mutant zebrafish *SOD1* and showed that this zebrafish model recapitulated the major phenotypes of ALS, including decreased endurance, neuromuscular junction defects, muscle pathology, and motor neuron loss [36].

FUS and TDP-43 are related with gene expression steps, such as microRNA processing, pre-mRNA splicing, and transcriptional regulation [103,104]. In addition, they shuttle between the cytoplasm and nucleus [105,106].

Mutations of FUS, an RNA-binding protein, are one of the causes of fALS, and more than 50 missense mutations reported so far are mainly located in exon 15, which encodes the nuclear localization signal in the C-terminal region of the protein. These mutations are known to cause FUS to redistribute into the cytoplasm for clearance from the nucleus [105,107,108]. To study *FUS* mutations, numerous model organisms have been generated and analyzed; however, several *Fus* knockout mice were reported to die within 16 h after birth. Affected cells showed increased aneuploidy and chromosomal aberrations [109]. In *Drosophila*, the gene is not responsible for the maintenance of adult neuronal function and may require additional contributions from FUS cytosolic aggregates and pathological changes in non-neuronal cell types [110]. Recently, researchers generated a deletion mutant for a unique *fus* ortholog in zebrafish. The findings showed a shortened lifespan and behavioral deficits with anatomical deficits, including neuromuscular junction disruption and shortened motor neuron length. Importantly, major motor deficits were rescued by overexpression of human FUS mRNA, but not by human TARDBP mRNA [37].

The *TARDBP* gene mutations are found in approximately 5% of fALS cases, 1% of sALS cases, and 1% of frontotemporal dementia (FTD) cases. Several model organisms with TDP-43 mutation have been generated from mouse to zebrafish; however, the mode by which TDP-43 mutation induces ALS pathophysiology remains poorly understood. Recently, researchers have generated a mutant TDP-43^G348C^ zebrafish model and observed several features, including the reduction of the locomotor activity, axonopathy of the motor neurons of the spinal cord, and branching defects in the secondary branches [111].

The most well-known genetic cause of ALS/FTD is a hexanucleotide extension within the first intron of the *C9orf72* gene [112,113]. Patients with *C9orf72* are known to have TDP-43 proteinopathy, but whether there is further crossover between *C9orf72* pathology and other ALS subtypes has yet to be revealed. Additionally, Shaw and colleagues generated a *C9orf72*-associated zebrafish model that stably expressed disrupted C4G2 amplification and exhibited RNA foci and dipeptide repeat protein (DPR) pathology. The zebrafish model had motor deficits, cognitive impairment, muscle atrophy, motor neuron loss, and mortality, as is likewise observed in early adult human ALS/FTD with *C9orf72* [39].

Zebrafish models utilizing various rare-disease-related genes described above are expected to be powerful tools for rapid drug screening and therapeutic development. Despite its importance as a biomedical model for rare disease studies, zebrafish possess some limitations, including the dissimilarity of some organs within the respiratory and reproductive systems. Thus, it is difficult to utilize zebrafish as a model for respiratory or reproductive contexts in humans. In some cases, the molecular mechanism between zebrafish and humans is not always unequivocal at the level of gene expression, protein modification, anatomical phenotype, physiology, or behavior. Moreover, the screening of water-insoluble drugs proves difficult, due to the aquatic habitat of zebrafish.

## 4. Conclusions

Until today, many researchers have conducted studies on rare disease by using model organisms, such as zebrafish. Although each model animal has its own set of advantages, we present the zebrafish as the ideal in vivo model to address this large pool of rare disease candidate genes. “Humanized” zebrafish have been advanced in the field of rare neurological disease research through various technologies, such as DNA editing technology (CRISPR/Cas9) and next-generation DNA sequencing.

As mentioned in Table 2, zebrafish are being utilized for disease model production and phenotype validation through candidate gene mutation in various rare neurological disease studies. Phenotypic features of mutant zebrafish are similar with human patients’ clinical and pathophysiologic features. Finally, we expect that, when the characteristics of zebrafish, genetic manipulation technology, and big data from next-generation DNA sequencing are combined, there will be great advances in understanding rare neurological diseases and developing diagnostic techniques and treatment approaches.

## Figures and Tables

**Figure 1 ijms-23-03946-f001:**
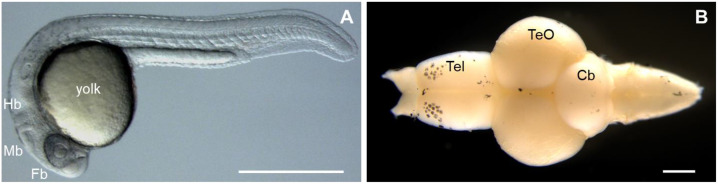
Major features of the zebrafish (*Danio rerio*). (**A**) Zebrafish embryo at day 1 of development. Fb, forebrain; Mb, midbrain; Hb, hindbrain. (**B**) Dissection of brain from an adult zebrafish. Tel, telencephalon; TeO, optic tectum; Cb, cerebellum. Rostral part of the brain is pointing to the left. Scale bars: 500 µm.

**Table 1 ijms-23-03946-t001:** List of rare disease research program or network.

Program/Network Name	Goals	Homepage Address
Genetic and Rare Diseases Information Center	Providing the general public with the latest information on various rare diseases in an easy-to-understand manner.	https://rarediseases.info.nih.gov/diseases
International rare disease research consortium	Contributing to the development of new treatments for rare diseases and methods to uncover the genetic causes of rare diseases.	https://irdirc.org/
National Organization for Rare Disorders	Raising awareness of rare diseases and improve access to treatment and medical services for patients and their families.	https://rarediseases.org
Orphanet (global network)	Providing international reference knowledge base for rare diseases and orphan drugs.	https://www.orpha.net/
	Providing the scientific datasets (research, clinical trials, drugs, etc.) related to rare diseases and orphan drugs.	http://www.orphadata.org/
Rare Diseases Clinical Research Network	Providing support for clinical studies and facilitating collaboration, study enrollment, and data sharing.	https://ncats.nih.gov/rdcrn
Undiagnosed Diseases Network	Accelerating identification of genetic causes of rare diseases by validating candidate genes, using model organisms.	https://undiagnosed.hms.harvard.edu/research

**Table 2 ijms-23-03946-t002:** Zebrafish modeling for rare neurological diseases.

Gene Name	Related Disease	Zebrafish Phenotype	Publication
*sod1*	Amyotrophic Lateral Sclerosis	Motor neuron loss Muscle atrophy	[36]
*fus*		Shortened motor neuron length Decreased neuromuscular junction Impaired motor behavior Decreased life span Increase of the smallest *tau* transcripts	[37]
*tardbp*		Axonopathy of the motor neurons Premature of axonal branch	[38]
*c9orf72*		Impaired motor behavior Cognitive impairment Muscle atrophy Motor neuron loss	[39]
*fam50a*	Armfield XLID syndrome	Abnormal neurogenesis Abnormal craniofacial patterning	[40]
*dyrk1a*	Down Syndrome and Autism	Decreased brain size Increased anxiolytic behavior Impaired social interaction/cohesion	[41]
*wdr11*	Idiopathic Hypogonadotropic Hypogonadism Kallmann Syndrome	Delayed puberty Impaired sense of smell	[42]
*eftud2*	Mandibulofacial Dysostosis, Guion–Almeida Type	Decreased brain size Small eyes Curved body Early embryonic lethality	[43]
*zc4h2*	Miles–Carpenter Syndrome	Abnormal swimming Increased twitching Motor hyperactivity Eye movement deficits Pectoral fin contractures	[44]
*phf21a*	Potocki–Shaffer Syndrome	Abnormal head and jaw size Change of head and face shape	[45]
*eif4a3*	Richieri–Costa–Pereira Syndrome	Underdevelopment of craniofacial cartilage and bone structures	[46]
*eif2b5*	Vanishing White Matter Disease	Early embryonic lethality Loss of oligodendrocyte precursor cells Impaired motor behavior	[47]
*eif2b3*		Defected myelin gene expression Defected glial cell differentiation	[48]
*sam2*	12q14.1 Deletion Syndrome	Increased of fear, anxiety-related behaviors, and autism	[49]

## Data Availability

Not applicable.

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
