# Peer review of "Disease Modeling of Rare Neurological Disorders in Zebrafish"

_ijms, 2022, doi:10.3390/ijms23073946_

Round 1
Reviewer 1 Report
The submitted manuscript is a review article describing the role of zebrafish in studying rare neurological diseases. I find it justified to take up this topic and give it some fresh point of view. However, I have some comments and suggestion that have to be considered before resubmission.
- The authors should reconsider the necessity of paragraphs describing C. elegans and fruit fly. They are superfluous and unlikely to add special value in the context of raised topic. Even more so they contain only one example of the study conducted in mentioned species. It is not enough to give any background.
- The main body of the text describing the use of zebrafish as a model is divided into several paragraphs describing individual diseases with almost the same construction – the description of the disease and one-two examples of the study conducted in zebrafish. These sections should compare and contrast published studies, identify gaps that have not been yet addressed. Review is a kind of theoretical discussion of relevant data on particular topic which should be critically analyzed. Submitted manuscript doesn’t share these features. Its construction is based on listing published studies step by step, without any comment or discussion.
- There is lack of general paragraph describing main features of zebrafish which make it a good model in the context of raised topic. There should be a short description of zebrafish nervous system (anatomical and physiological) pointing the similarities between zebrafish and human. Also there is a need of listing and comment main techniques which are used in investigations of rare neurological diseases based on zebrafish model.
- Finally, I find it necessary to indicate the limitations of the zebrafish usage in such researches.
Author Response
Response to Reviewer 1’s Comments
The submitted manuscript is a review article describing the role of zebrafish in studying rare neurological diseases. I find it justified to take up this topic and give it some fresh point of view. However, I have some comments and suggestion that have to be considered before resubmission.
Point 1: The authors should reconsider the necessity of paragraphs describing C. elegans and fruit fly. They are superfluous and unlikely to add special value in the context of raised topic. Even more so they contain only one example of the study conducted in mentioned species. It is not enough to give any background.
Response 1: We appreciate the suggestion and have removed the case study of C. elegans and fruit fly for easier understanding by the reader. We wanted, however, to keep its introductory paragraphs and have made the following changes below.
|
New version: Section 2.1. Caenorhabditis elegans (C. elegans) The significance of non-mammalian model organisms has been recognized for quite some time [1, 6]. The Nobel Prize in Physiology or Medicine has been awarded to researchers for their discoveries in apoptosis (2022) and RNA interference using worms (2006). C. elegans is an unsegmented pseudocoelomate, lacking circulatory and respiratory systems, whose body segments are serially repeated one after the other [7]. C. elegans was used primarily for neuronal development research in 1963 and has since seen extensive use as a model organism for research into neural and molecular mechanisms of learning, memory, mating behavior, chemotaxis, thermotaxis, and mechano-transduction [8]. In addition, the C. elegans model has provided insights into finer mechanistic details of human health including inter-cellular signaling pathways (e.g. Notch signaling), intra-cellular pathways (e.g. autophagy), molecular machines (e.g. spliceosome), and multi-cellular processes (e.g. basement membrane biology) [9]. In 2019, the first C. elegans multicellular organism underwent whole-genome sequencing and is the only organism to have completed its connectome (the "wiring diagram" of neurons) [10-12]. There are 20,512 protein-coding genes [13] and 83% of the worm proteome is found to have human homologous genes [14]. Only 11% or less contains roundworms specific genes. These findings provide the basis by which C. elegans has served as a suitable model organism for human gene functional research [15-18]. In the case of rare disease modeling, C. elegans provides an ideal system to study human diseases. Spinal muscular atrophy (SMA) is an inherited neuromuscular disease in which the loss of motor neurons in the spinal cord causes muscle weakness and inability to control muscle movements in areas such as the head, neck, legs and arms. About 25,000 adults and children live with SMA in the United States. Survival motor neuron (SMN) protein is encoded by the SMN1 gene in human and SMA is caused by mutated or missing two copies of SMN1 gene. SMA research studying the neuronal calcium sensor Neurocalcin delta (NCALD) as a protective modifier was conducted in C. elegans. The NCALD is a calcium dependent negative regulator of endocytosis, and its knockdown effectively ameliorates SMA-associated pathological defects. Interestingly, these results, corresponded with findings across species, including worm, zebrafish, and mouse [18]. They demonstrate the utility of C. elegans for physiological evaluation of genetic relationships using well-established tools such as RNAi or publicly available mutant strains.
New version: Section 2.2. Fruit fly (Drosophila melanogaster) Drosophila melanogaster is a species of fly in the Drosophila family (taxonomic order Diptera). For the past century, the Drosophila melanogaster has been used as a model organism to understand the fundamentals of genetics, developmental biology, immunity, and neuroscience [19, 20]. Five Nobel Prizes have been awarded to fruit fly scientists for their work with the animal in 2017. Drosophila melanogaster has emerged as an important model system for dissecting and understanding the molecular mechanisms underlying human rare diseases due to its rapid life cycle, relatively simple genetics (only four pairs of chromosomes), and a large number of offspring per generation. This rise is owed in part to the first genome-wide survey of ~1,000 genes registered in the Online Mendelian Inheritance in Man (OMIM) which found that 75% of disease-causing genes in humans are conserved in Drosophila [21]. Of the approximately 4,000 human disease-related genes currently displayed in OMIM, ~85% have homologues in Drosophila. Considering that ~65% of protein-coding genes are conserved between humans and flies, the data suggest that genes conserved between these species are more likely to implicate genetic disease in humans [22, 23]. In addition to being used as a tool for dissecting rare disease mechanisms and exploring potential therapeutic avenues, flies have emerged as a key tool in explaining variants of uncertain significance found in patients [24, 25]. For example, human congenital microcephaly is characterized by a reduction in brain size. Link and colleagues showed that Ankryin repeat and LEM domain 2 (ANKLE2), an endoplasmic reticulum (ER)-localized protein, is critical for proper ER and nuclear envelope structure. The Ankle2 gene mutation disrupts ER and nuclear envelope morphology, releasing kinase Ballchen-VRK1 into the cytoplasm. These defects were associated with decreased aPKC phosphorylation, disrupted Par complex localization, and defective spindle alignment. In addition, 3 variants of the ANKLE2 mutation were identified in six microcephaly patients. As seen in humans, this mutation reduced brain volume and affected the function of Par complex in Drosophila. Expression of NS4A, which binds to and inhibits the function of Ankle2, phenocopied Ankle2 mutant defects in neuroblasts. These defects were subsequently rescued by modulation of the Ankle2 pathway [25].
|
Point 2: The main body of the text describing the use of zebrafish as a model is divided into several paragraphs describing individual diseases with almost the same construction – the description of the disease and one-two examples of the study conducted in zebrafish. These sections should compare and contrast published studies, identify gaps that have not been yet addressed. Review is a kind of theoretical discussion of relevant data on particular topic which should be critically analyzed. Submitted manuscript doesn’t share these features. Its construction is based on listing published studies step by step, without any comment or discussion.
Response 2: We thank the reviewer for this comment. In this review paper, a rare neurologic disease model was produced using zebrafish and its characteristics (phenotypes, behaviour etc.) were detailed. In addition, examples of various genetic diseases and target (cause) genes were shown to clarify the reason for selecting zebrafish over other model organisms such as the fruit fly or C. elegans. As pointed out by the reviewer, this review paper needs to compare and contrast published studies, but we wanted to focus on a series of processes to find the causes of various genetic rare diseases and select the causative gene candidates for rare diseases using zebrafish modelling. Because many of these findings in rare diseases are with novel genes and are at very early stage of mechanism study, it is not easy to compare them directly with other rare diseases.
Point 3-1: There is lack of general paragraph describing main features of zebrafish which make it a good model in the context of raised topic. There should be a short description of zebrafish nervous system (anatomical and physiological) pointing the similarities between zebrafish and human.
Response 3-1: We appreciate the reviewer’s comment. As suggested, a short description of the anatomical and physiological zebrafish nervous system has been added. The changes are as follows below.
|
New version: Section 2.3. Zebrafish (Danio rerio) Interestingly, similarities between the zebrafish and human nervous system (anatomy and physiological signaling) have been reported [33]. The zebrafish brain is composed of the forebrain, midbrain, and hindbrain (Fig. 1D); and many cells including astrocytes, oligodendrocytes, microglia, cerebellar Purkinje cells, myelin and motor neurons are also similar to human cells. Further studies of spinal nerve patterning, neural differentiation, and vertebrate network development in adult zebrafish revealed similarities to higher vertebrates [33]. Due to these characteristics, zebrafish are widely used to validate candidate disease genes and to elucidate the molecular mechanisms and pathophysiology of neurological disease. The continued increase in the use of zebrafish in biomedical research publications reflects their expanding popularity [33] (Table 2).
|
Point 3-2: There is a need of listing and comment main techniques which are used in investigations of rare neurological diseases based on zebrafish model.
Response 3-2: We appreciate the reviewer’s comment. Accordingly, we added information on genetic technologies for rare neurological diseases based on the zebrafish model. Advanced gene-targeting technologies using ZFNs, TALENs, and CRISPR/Cas9 have overcome the challenges of generating specific gene knockout mutations due to improved resolution of the zebrafish genome map​​. Among them, we focused more on the CRISPR/Cas9 system. The CRISPR/Cas9 system relies on target site recognition by a custom guide RNA (gRNA) molecule and requires only one oligonucleotide to be designed for each target site. The success of CRISPR-mediated transgenic transformation in zebrafish can largely be attributed to frame-shift-generating null alleles via non-homologous extremity joint (NHEJ)-mediated CRISPR-induced DNA break repair. The respective sentences have been added as follows below.
|
New version: Section 2.3. Zebrafish (Danio rerio) The zebrafish possesses many characteristics that make it an invaluable model to study human diseases [27]; however, one of its unique advantages is the unparalleled optical clarity of the embryo, which allows visualization of individual genes (fluorescently stained or labeled) throughout development using non-invasive imaging techniques [28-31] (Fig. 1C). This transparency of the embryo also facilitates genetic manipulation such that gene function can readily be studied by the injection of synthetic mRNA or plasmid DNA into early-stage zebrafish embryos generating transgenic zebrafish lines or altering gene function through genome editing techniques such as the including zinc finger nucleases (ZFNs), transcription activation-like effector nucleases (TALENs), and the clustered regularly interspaced repeat (CRISPR)/CRISPR-associated protein 9 (Cas9) system [32-34]. Furthermore, the zebrafish genome has been sequenced and 71% of human genes and 82% of human disease-related genes have orthologs in zebrafish [35]. Both ZFNs and TALENs require the creation of customized protein compositions for each target site, and these systems are not suitable for large-scale applications. However, the CRISPR/Cas9 system relies on target site recognition by a custom guide RNA (gRNA) molecule and requires only one oligonucleotide to be designed for each target site. The success of CRISPR-mediated transgenic transformation in zebrafish can largely be attributed to frame-shift-generating null alleles via non-homologous extremity joint (NHEJ)-mediated CRISPR-induced DNA break repair [35].
|
Point 4: Finally, I find it necessary to indicate the limitations of the zebrafish usage in such researches.
Response 4: We thank the reviewer for this comment. Accordingly, we have added the following sentences and reference regarding zebrafish limitations, as seen below.
|
New version: Section 3.9. Amyotrophic Lateral Sclerosis (Lou Gehrig’s disease) Zebrafish models utilizing various rare disease-related genes described above are expected to be powerful tools for rapid drug screening and therapeutic development. Despite the various characteristics, the limitations of the zebrafish as a model organism are clearly present. About 20 percent of zebrafish genes have two functional copies due to additional duplications throughout the genome compared to mammals, complicating the generation of knockout disease models [116]. References 116. Ali, S.; Champagne, D. L.; Spaink, H. P.; Richardson, M. K. Zebrafish embryos and larvae: a new generation of disease models and drug screens. Birth Defects Res. C Embryo Today, 2011, 93, 115–133.
|

Reviewer 2 Report
A good review that contains great information not reviewed elsewhere. This is useful information to synthesize for the rare disorder research community, and will likely assist researchers working on these conditions to develop effective research strategies.
The article is well written and doesn't leave the reader wanting for more information. It seems to cover all of the information that would be sensible when writing a review on this topic.
The two tables are very useful information, although Figure 1 doesn't add much value since it's just pictures of zebrafish. However, the value could be increased by referencing the figure more thoroughly in the text to demonstrate why the images in each panel are relevant. For example, Panel 1B could be cited on page 4 lines 156-158 when discussing optical clarity of the embryo since panel 1B illustrates to the reader the optical clarity. Panel 1C can be referenced in lines 147 to 149 to illustrate the potential for high-throughput screening. The brain image in 1D is not really mentioned anywhere so it might be good to add a sentence or two explaining how zebrafish are good models for neurological and behavioral studies. Despite the fact that the rest of the review is dedicated to great examples of that, it would be good to reference the brain image somewhere to give the figure some illustrative value.
Author Response
Response to Reviewer 2’s Comments
A good review that contains great information not reviewed elsewhere. This is useful information to synthesize for the rare disorder research community, and will likely assist researchers working on these conditions to develop effective research strategies.
The article is well written and doesn't leave the reader wanting for more information. It seems to cover all of the information that would be sensible when writing a review on this topic.
Point 1: The two tables are very useful information, although Figure 1 doesn't add much value since it's just pictures of zebrafish. However, the value could be increased by referencing the figure more thoroughly in the text to demonstrate why the images in each panel are relevant. Panel 1B could be cited on page 4 lines 156-158 when discussing optical clarity of the embryo since panel 1B illustrates to the reader the optical clarity.
Response 1: We appreciate the reviewer’s comment. The order of figures has been rearranged to help reader understanding. Figure 1B (embryo image) was changed to Figure 1C and is cited on page 4 lines 156-158. Figure 1 and its legend were revised in manuscript and the changes can be seen below.
|
New version: Figure 1 and its legend
Figure 1. Major features of the zebrafish (Danio rerio). (A) A group of zebrafish swimming in an aquarium. (B) Zebrafish larvae in a 96-well microplate. (C) Zebrafish embryo at day 1 of development. (D) Dissection of brain from an adult zebrafish. Pictures not to scale. New version: Section 2.3. Zebrafish (Danio rerio) The zebrafish possesses many characteristics that make it an invaluable model to study human diseases [27]; however, one of its unique advantages is the unparalleled optical clarity of the embryo, which allows visualization of individual genes (fluorescently stained or labeled) throughout development using non-invasive imaging techniques [28-31] (Fig. 1C).
|
Point 2: Panel 1C can be referenced in lines 147 to 149 to illustrate the potential for high-throughput screening.
Response 2: We appreciate the reviewer comment. The order of figures has been rearranged to help with reader understanding. Figure 1C (zebrafish larvae in multi-well plate image) was changed to Figure 1B and is cited on page 4 lines 147-149. Figure 1 and its legend were revised in manuscript and the changes can be seen below.
|
New version: Figure 1 and its legend
Figure 1. Major features of the zebrafish (Danio rerio). (A) A group of zebrafish swimming in an aquarium. (B) Zebrafish larvae in a 96-well microplate. (C) Zebrafish embryo at day 1 of development. (D) Dissection of brain from an adult zebrafish. Pictures not to scale. New version: Section 2.3. Zebrafish (Danio rerio) In addition, zebrafish have large clutch sizes ranging from 200 to 300 embryos per adult mating pair which ensures a robust stock of animals for research work. Due to these features combined with the relatively small size of the embryo/larva/adult, zebrafish are well-suited toward high-throughput screening of potential neuroactive compounds (Fig. 1B).
|
Point 3: The brain image in 1D is not really mentioned anywhere so it might be good to add a sentence or two explaining how zebrafish are good models for neurological and behavioral studies. Despite the fact that the rest of the review is dedicated to great examples of that, it would be good to reference the brain image somewhere to give the figure some illustrative value.
Response 3: We appreciate and agree with this comment. In order to demonstrate that the zebrafish is a good model for neurological and behavioural research, we first added a description of the zebrafish nervous system (anatomical and physiological) followed by examples of similarity between humans and zebrafish. Figure 1D was also cited on page 5 lines 166.
|
New version: Section 2.3. Zebrafish (Danio rerio) Interestingly, similarities between the zebrafish and human nervous system (anatomy and physiological signaling) have been reported [33]. The zebrafish brain is composed of the forebrain, midbrain, and hindbrain (Fig. 1D); and many cells including astrocytes, oligodendrocytes, microglia, cerebellar Purkinje cells, myelin and motor neurons are also similar to human cells. Further studies of spinal nerve patterning, neural differentiation, and vertebrate network development in adult zebrafish revealed similarities to higher vertebrates [33]. Due to these characteristics, zebrafish are widely used to validate candidate disease genes and to elucidate the molecular mechanisms and pathophysiology of neurological disease. The continued increase in the use of zebrafish in biomedical research publications reflects their expanding popularity [33] (Table 2).
|

Round 2
Reviewer 1 Report
I would like to thank the Authors for their reconsiderations and taking into account the suggestions.
- Despite the fact that the manuscript still doesn’t have the form of well written review, I accept the explanations why it has such construction.
- I would recommend to exclude or change the pictures. They are small and seem to not have a value in the context of the study. The more that figures A and B aren’t even cited or commented anywhere. The picture of ZF brain is relevant, however not in that form. It should be bigger, maybe the addition of some drawing/sketch with the description, where hindbrain, midbrain, forebrain etc borders are present would be a good idea. I also recommend to compare the drawings of ZF and human brain to indicate the similarities and differences. The transparency of the embryo is important in the context of raised issue as well is in the context of different studies, however it is a little bit obvious feature.
- The limitation of zebrafish usage is not only gene duplication. Please also indicate the differences in nervous system anatomy and physiology. Also looking at the zebrafish as a model for widely understood diseases we have to remember that in zebrafish we are able to investigate single mechanisms not all the aspects of the disease. For example, investigating something even at the molecular level is not always unequivocal with the protein level, phenotype, behavior etc.
Author Response
POINT BY POINT RESPONSES:
- “Despite the fact that the manuscript still doesn’t have the form of well written review, I accept the explanations why it has such construction.”
Response: Thank you very much.
- “I would recommend to exclude or change the pictures. They are small and seem to not have a value in the context of the study. The more that figures A and B aren’t even cited or commented anywhere. The picture of ZF brain is relevant, however not in that form. It should be bigger, maybe the addition of some drawing/sketch with the description, where hindbrain, midbrain, forebrain etc borders are present would be a good idea. I also recommend to compare the drawings of ZF and human brain to indicate the similarities and differences. The transparency of the embryo is important in the context of raised issue as well is in the context of different studies, however it is a little bit obvious feature.”
Response: We appreciate the recommendation and have removed Figures A and B. Instead, we had made Figures A and B bigger and added marks for forebrain (telencephalon), midbrain (optic tectum), and hindbrain (cerebellum).
- “The limitation of zebrafish usage is not only gene duplication. Please also indicate the differences in nervous system anatomy and physiology. Also looking at the zebrafish as a model for widely understood diseases we have to remember that in zebrafish we are able to investigate single mechanisms not all the aspects of the disease. For example, investigating something even at the molecular level is not always unequivocal with the protein level, phenotype, behavior etc.”
Response: Thank you for pointing this out. We have changed it as “Despite its importance as a biomedical model for rare disease studies, zebrafish possess some limitations, including the dissimilarity of some organs within the respiratory and reproductive systems. Thus, it is difficult to utilize zebrafish as a model for respiratory or reproductive contexts in humans. In some cases, the molecular mechanism between zebrafish and humans is not always unequivocal at the level of gene expression, protein modification, anatomical phenotype, physiology, or behavior. And, the screening of water insoluble drugs proves difficult due to the aquatic habitat of zebrafish.”
